# Dissipative cooling induced by pulse perturbations

**Andrea Nava[1, 2]⋆ and Michele Fabrizio[3]**

**1** Dipartimento di Fisica, Università della Calabria,
Arcavacata di Rende I-87036, Cosenza, Italy
**2** INFN - Gruppo collegato di Cosenza, Arcavacata di Rende I-87036, Cosenza, Italy
**3** International School for Advanced Studies (SISSA), Via Bonomea 265, I-34136 Trieste, Italy

⋆ andrea.nava@fis.unical.it

## Abstract

We investigate the dynamics brought on by an impulse perturbation in two infinite-range quantum Ising models coupled to each other and to a dissipative bath. We show that, if dissipation is faster the higher the excitation energy, the pulse perturbation cools down the low-energy sector of the system, at the expense of the high-energy one, eventually stabilising a transient symmetry-broken state at temperatures higher than the equilibrium critical one. Such non-thermal quasi-steady state may survive for quite a long time after the pulse, if the latter is properly tailored.



## 1 Introduction

Shooting ultrashort laser pulses has emerged in the last decades as a new fast-driving tool for phase transformations, with great potentials especially for strongly correlated materials [1–

9], whose phase diagrams include close-by insulating, conducting and even superconducting phases. Moreover, exciting a strongly correlated material by a laser pulse not always boils down to a fast rise of the internal temperature, as one would reasonably expect. For instance, it sometimes allows uncovering hidden states inaccessible at thermal equilibrium [10, 11]. However, until now the most remarkable failure of the naïve correspondence between light firing and thermal heating is the evidence of superconducting-like behaviour at nominal temperatures far higher than the critical one in the molecular conductors $K_3C_{60}$ and $\kappa$-(BEDT-TTF)$_2$Cu[N(CN)$_2$]Br irradiated by laser pulses [12–14]. Such non-thermal state is transient, but may become rather long-lived by properly tailoring the laser pulse [14]. Even though this phenomenon may have explanations that concern material-specific mechanisms [12,13], still it is legitimate to address the general question whether a laser pulse could ever cool down a solid state material. Spontaneous anti-Stokes emission of photons with higher energies than those absorbed from the incident light is a known laser cooling mechanism for semiconductors [15–19]. However, the same mechanism would not work in metals at low temperatures, e.g., in the above mentioned molecular conductors, where most of the entropy is carried by the electrons, and not by the phonons as in semiconductors.

In an attempt to explain the photoinduced superconductivity in $K_3C_{60}$ [12], a different laser cooling mechanism was proposed in Ref. [20], which is essentially based on the existence of a high energy localised mode that, when the laser is on, is able to fast absorb entropy from the thermal bath of low-energy particle-hole excitations, while, after the end of the laser pulse, it release back that absorbed entropy very slowly. It follows that the population of particle-hole excitations gets reduced, as if its internal temperature were lower, for a transient time after the pulse that is longer the smaller the non-radiative decay rate of the high energy mode. This idea was later tested [21] with success in a fully-connected toy model subject to a time dependent perturbation of finite duration, mimicking a 'laser pulse'. This model is trivially solvable since infinite connectivity implies that mean-field theory is exact in the thermodynamic limit. However, this feature, though providing the exact out-of-equilibrium dynamics, yet prevents full thermalisation, since it lacks internal dissipation. Therefore, despite the model does realise the laser cooling mechanism proposed in [20], one cannot exclude that dissipation could wash it out.

The aim of the present work is just to assess the role of dissipation in that same model. Since we want to maintain its mean-field character, we still assume full connectivity. Therefore dissipation cannot arise from the internal degrees of freedom, but it is simple included in the dynamics via a Lindblad equation. When all system excitations dissipate equally fast, we find just a quick relaxation to thermal equilibrium. On the contrary, if excitations dissipate faster the higher the energy, which is the most common physical situation, we do observe a transient regime where the low energy sector of the model effectively cools down. Moreover, if we increase the 'laser pulse' duration keeping constant the total energy supplied to the system, following the experiment in [14], we also find the transient state to last longer, not in disagreement with that experiment.

The paper is organized as follows. In Section 2 we introduce the quantum Ising model we shall investigate, an effective two-spin spin boson model [22] and its behaviour in absence of dissipation. In Section 3 we briefly discuss the Lindblad master equation to describe the relaxation dynamic and the physical results obtained for different bath model. Finally, Sec. 4 is devoted to concluding remarks.

## 2  The model Hamiltonian

We consider the Hamiltonian of two coupled fully-connected quantum Ising models

$$\hat{H} = \sum_{n=1}^{2} \hat{H}_n - \lambda \sum_{j=1}^{N} \sigma_{1,j}^x \sigma_{2,j}^x \,, \tag{1}$$

where

$$\hat{H}_n = -\frac{J}{2N} \sum_{i,j=1}^{N} \sigma_{n,i}^x \sigma_{n,j}^x - h_n \sum_{i=1}^{N} \sigma_{n,i}^z \,, \tag{2}$$

and $\sigma_{n,i}^\alpha$, $\alpha = x, y, z$, are Pauli matrices on site $i = 1, \ldots, N$ of the submodel $n = 1, 2$. All parameters, $J$, $h_1$, $h_2$ and $\lambda$ are assumed positive. Hereafter we take $J = 1$ as energy unit. Because of full connectivity, and for any $i \neq j$,

$$\left\langle \sigma_{n,i}^\alpha \sigma_{m,j}^\beta \right\rangle - \left\langle \sigma_{n,i}^\alpha \right\rangle \left\langle \sigma_{n,j}^\beta \right\rangle \propto \frac{1}{N} \,, \tag{3}$$

which actually implies that the mean-field approximation becomes exact in the thermodynamic limit $N \to \infty$, or, equivalently, that the full density matrix $\hat{\rho}$ factorises in that limit into the product of single-site density matrices:

$$\hat{\rho} \quad \xrightarrow[N \to \infty]{} \quad \prod_{i=1}^{N} \hat{\rho}_i \,, \tag{4}$$

where $\hat{\rho}_i$ are positive definite $4 \times 4$ matrices with unit trace. The property (4) allows exactly solving with relative ease the model Hamiltonian (1).

### 2.1  Equilibrium phase diagram

At equilibrium, one can exploit the variational principle for the free energy at temperature $T$,

$$F(T) = \min_{\hat{\rho}} \left[ \mathrm{Tr}\left( \hat{\rho} \hat{H} \right) + T \, \mathrm{Tr}\left( \hat{\rho} \ln \hat{\rho} \right) \right], \tag{5}$$

to find, through Eq. (4), the single-site density matrices $\hat{\rho}_i^{(eq)}(T)$ that minimise the r.h.s. of Eq. (5), and which actually solve the self-consistency set of equations

$$\hat{\rho}_i^{(eq)}(T) = \frac{e^{-\beta \hat{H}_i(T)}}{\mathrm{Tr}\left( e^{-\beta \hat{H}_i(T)} \right)} \,,$$

$$\hat{H}_i(T) = -\sum_{n=1}^{2} \left[ J_{n,i}^{(eq)}(T) \sigma_{n,i}^x + h_n \sigma_{n,i}^z \right] - \lambda \sigma_{1,i}^x \sigma_{2,i}^x \tag{6}$$

$$\equiv \sum_{n=1}^{2} \hat{H}_{n,i} - \lambda \sigma_{1,i}^x \sigma_{2,i}^x \,,$$

where

$$J_{n,i}^{(eq)}(T) = \lim_{N \to \infty} \frac{1}{N} \sum_{j=1}^{N} \mathrm{Tr}\left( \hat{\rho}_j^{(eq)}(T) \sigma_{n,j}^x \right) \equiv J_n^{(eq)}(T). \tag{7}$$

Since $J_{n,i}^{(eq)}(T) = J_n^{(eq)}(T)$ is the same for all sites, so $\hat{\rho}_i^{(eq)}(T)$ and $\hat{H}_i(T)$ in (6) are. Therefore, we can also write

$$J_{n,i}^{(eq)}(T) = \mathrm{Tr}\left( \hat{\rho}_i^{(eq)}(T) \sigma_{n,i}^x \right) \equiv m_{x,n}(T), \quad \forall i, \tag{8}$$

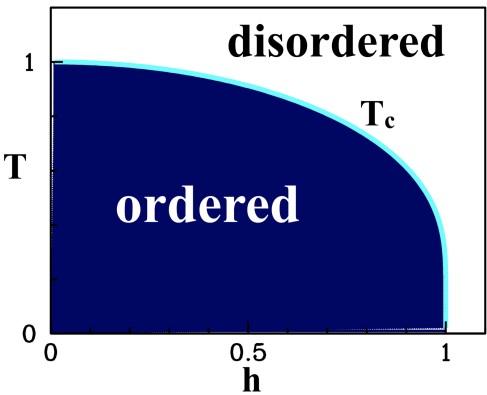

Figure 1: Phase diagram of the fully connected quantum Ising model (2) as a function of the transverse field $h$. In the blue coloured region, below the critical temperature, the system is in the ordered phase with spontaneously broken $Z_2$ symmetry.

which, together with Eq. (6), give an equivalent representation of the self-consistency equations.

At $\lambda = 0$ in Eq. (1), each Ising model (2) has the phase diagram shown in Fig. 1. For $h_n \leq 1$ and temperature

$$T \leq T_c(h_n) = \frac{2h_n}{\ln \dfrac{1+h_n}{1-h_n}} \,, \tag{9}$$

the expectation value $m_{x,n}(T)$ of $\sigma^x_{n,i}$, see Eq. (8), is finite, thus the model $n$ is in an ordered phase that spontaneously breaks the $Z_2$ symmetry $\sigma^x_{n,i} \to -\sigma^x_{n,i}$, $\forall i$. Above $T_c(h_n)$ or if $h_n > 1$, the $Z_2$ symmetry is restored, and the order parameter $m_{x,n}(T)$ vanishes identically. The mean-field Hamiltonian $\hat{H}_{n,i}$ of each subsystem $n = 1, 2$, see Eq. (6), has two eigenstates separated by an energy

$$E_n(T) = 2 \sqrt{ m_{x,n}(T)^2 + h_n^2 } \,, \tag{10}$$

which corresponds to a dispersionless optical excitation branch of the Hamiltonian $\hat{H}_n$ in Eq. (2). For $h_n \leq 1$, $E_n(0) = 2$ at $T = 0$, and diminishes with $T$ until, at $T = T_c(h_n)$ and above, $E_n(T) = 2h_n$.

Throughout this work we assume

$$\lambda \ll h_1 < 1 \ll h_2 \,, \tag{11}$$

and, specifically,

$$\lambda = 10^{-2}, \quad h_1 = 0.5, \quad h_2 = 10. \tag{12}$$

In this case, for $T \leq T_c \simeq T_c(h_1)$, model 1 acquires a finite order parameter $m_{x,1}(T)$, which, in turn, drives a finite

$$m_{x,2}(T) \simeq \frac{\lambda}{h_2} \, m_{x,1}(T) \ll m_{x,1}(T) \,. \tag{13}$$

It follows that the Hamiltonian (1) has, at leading order in $\lambda$, two dispersionless excitation branches with energies

$$E_1(T) \simeq 2 \sqrt{ m_{x,1}(T)^2 + h_1^2 } \,,$$
$$E_2(T) \simeq 2h_2 \gg \Delta E_1(T) \,. \tag{14}$$

In other words, we can write

$$\hat{H} \simeq \sum_{i=1}^{N} \sum_{n=1}^{2} E_n(T) \, b_{n,i}^{\dagger} \, b_{n,i} \,, \tag{15}$$

with $b_{n,i}$ hard core bosons. The local Hilbert space at site $i$ thus comprises four eigenstates

$$|k;i\rangle \equiv \left(b_{1,i}^{\dagger}\right)^{n_1(k)} \left(b_{2,i}^{\dagger}\right)^{n_2(k)} |0\rangle \,, \quad k = 0,\dots,3 \,, \tag{16}$$

with energy

$$E(k) = n_1(k) E_1(T) + n_2(k) E_2(T) \,, \tag{17}$$

where $n_2(k) = \lfloor k/2 \rfloor$ is the integer part of $k/2$, and $n_1(k) = k - 2n_2(k)$. The states $|0;i\rangle$ and $|1;i\rangle$ define a low energy subspace well separated from the high energy one, which includes states $|2;i\rangle$ and $|3;i\rangle$. It follows that there exists a wide temperature interval, $T_c \lesssim T \ll E_2(T) = 2h_2$, where the low energy sector is entropy rich, contrary to the high energy one, which practically bears no entropy.

## 2.2 Cooling strategy

Based on the last observation, Ref. [21] devised a strategy to exploit the high energy sector as an entropy sink able to cool down the low energy one, which we briefly sketch in this section. We assume that the system is prepared in the equilibrium state corresponding to a temperature $T_c \lesssim T \ll 2h_2$. Its initial density matrix $\hat{\rho}(0)$ is therefore defined through Eq. (4), with $\hat{\rho}_i$ the solution of the self consistency equations (6) and (8). At $t = 0$ the following perturbation is turned on

$$
\begin{aligned}
\hat{V}(t) &= -E(t) \cos \omega t \, \sum_{i=1}^{N} \sigma_{1,i}^{x} \sigma_{2,i}^{x} \\
&\approx -E(t) \cos \omega t \, \sum_{i=1}^{N} \left(b_{1,i}^{\dagger} + b_{1,i}\right) \left(b_{2,i}^{\dagger} + b_{2,i}\right) \,,
\end{aligned}
\tag{18}
$$

which mimics a laser pulse, whose envelope we hereafter parametrise as

$$E(t) = \left(\frac{t}{\tau}\right)^2 \exp\left[1 - \frac{1}{E_0}\left(\frac{t}{\tau}\right)^2\right] \,, \tag{19}$$

thus corresponding to a pulse of duration $\tau$ and peak amplitude $E_0$ achieved for $t_{max} = \tau \sqrt{E_0}$. The perturbation (18) allows for the transition processes $b_{2,i}^{\dagger} b_{1,i} = |2;i\rangle\langle 1;i|$ and $b_{2,i}^{\dagger} b_{1,i}^{\dagger} = |3;i\rangle\langle 0;i|$ plus their hermitean conjugates (as depicted in panel a) of Fig. 3). It is worth to stress that, in our model spins and, equivalently, hard core bosons, are just a tool to reproduce a Hilbert space comprising four energy states for each site, two, $|0;i\rangle$ and $|1;i\rangle$, with lower energy and two, $|2;i\rangle$ and $|3;i\rangle$, with higher one. The laser pulse allows transitions between the two sectors, specifically, $0 \leftrightarrow 3$ and $1 \leftrightarrow 2$, which are simply interpreted in terms of hard-core bosons. The key to our mechanism is to tune the laser frequency in resonance with $1 \leftrightarrow 2$, thus depleting the occupation of boson $b_{1,i}$ and increasing that of $b_{2,i}$. In this sense (19) mimics a laser pulse, acting at the level of single particles. In analogy with optical absorption of light we can image that $b_{1,i}$ is even under parity and $b_{2,i}$ odd, so that all transition processes $0 \leftrightarrow 3$ and $1 \leftrightarrow 2$ are dipole active, i.e. couple opposite parity states. We further assume the 'laser' frequency $\omega = E_2(T) - E_1(T)$, see Eq. (14), in resonance with the excitation process $b_{2,i}^{\dagger} b_{1,i} = |2;i\rangle\langle 1;i|$, and the hermitean conjugate de-excitation one. The rationale behind this choice is the following. If $p_k(T) = \langle |k;i\rangle\langle k;i| \rangle$ is the initial

occupation probability, i.e. the equilibrium one at temperature $T$, of the state $|k;i\rangle$, then, for $T_c \lesssim T \ll 2h_2$, the high-energy sector starts almost unoccupied, $p_2(T) \simeq p_3(T) \simeq 0$, while

$$1 \geq \frac{p_1(T)}{p_0(T)} = \frac{1 - \tanh \beta \, h_1}{1 + \tanh \beta \, h_1} \gtrsim \frac{p_1(T_c)}{p_0(T_c)} = h_1 \,. \tag{20}$$

The effect of the 'laser pulse' (18) is primarily to increase the formerly negligible $p_2$ by reducing $p_1$, eventually making the ratio $p_1/p_0$ drop beneath the threshold value $h_1$, below which $Z_2$ symmetry breaking, which is mostly a matter of the lower energy sector, spontaneously sets in. In other words, the system starts in the disordered phase above $T_c$ and, after the 'laser pulse', it may end up into the ordered one, as if it were cooler than it was initially. In reality, the energy lost by the low-energy sector plus that soaked up from the 'laser pulse' is being temporarily stored in the high-energy sector, from which it later flows back and heats up the whole system, though gradually since $\lambda$ is tiny. Therefore, the low energy sector is transiently emptied of entropy by the 'laser pulse', for a time longer the small $\lambda$ is.

This scenario, put forth in Ref. [21], can be readily shown to occur in the model Hamiltonian (1) in presence of the perturbation (18). Indeed, since the full time-dependent Hamiltonian $\hat{H}(t) = \hat{H} + \hat{V}(t)$ does not invalidate Eq. (3), the time-dependent density matrix $\hat{\rho}(t)$ can be still written as in Eq. (4) with $\hat{\rho}_i \to \hat{\rho}_i(t)$, where the latter evolves according to the first order non-linear differential equation,

$$\frac{\partial \hat{\rho}_i(t)}{\partial t} = -i \left[ \hat{H}_i(t), \hat{\rho}_i(t) \right], \tag{21}$$

where, similarly to Eqs. (6) and (8),

$$\hat{H}_i(t) = -\sum_{n=1}^{2} \left[ J_{n,i}(t) \, \sigma_{n,i}^x + h_n \, \sigma_{n,i}^z \right] - \left( \lambda + E(t) \cos \omega t \right) \sigma_{1,i}^x \sigma_{2,i}^x \,, \tag{22}$$

and the non-linearity arises because

$$J_{n,i}(t) = \mathrm{Tr}\left( \hat{\rho}_i(t) \, \sigma_{n,i}^x \right), \tag{23}$$

is function of $\hat{\rho}_i(t)$. Eq. (21) must be solved with initial condition $\hat{\rho}_i(t=0) = \hat{\rho}_i^{(eq)}(T)$, which, being actually the same for all sites $i$, implies that also $\hat{\rho}_i(t>0)$ is site-independent. Therefore one just needs to solve (21) for a single-site.

In Fig. 2 we show the time evolution of the low-energy sector order parameter $m_{x,1}(t)$ starting from the disordered equilibrium phase at $T = 1.5 \, T_c$, and using, besides the Hamiltonian parameters in Eq. (12), a 'laser pulse' of duration $\tau = 250$ and amplitude $E_0 = 0.20$, see Eq. (19). We note that initially $m_{x,1} = 0$, since the system is disordered. However, after the 'laser pulse' the low energy sector ends up trapped into one of the two $Z_2$-equivalent symmetry variant phases, in the figure that with $m_{x,1}$ negative.

## 3 Dissipative dynamics

We already mentioned that the integrability of the Hamiltonian (1) has as counterpart the lack of any internal dissipation, as evident by the undamped oscillations in Fig. 2. This evidently raises doubts about the general validity of the results in the previous section. We could add internal dissipation giving up the possibility of exactly solving the model, e.g., by defining it on a lattice, and making the exchange $J$ in Eq. (1) decaying with the lattice distance between two

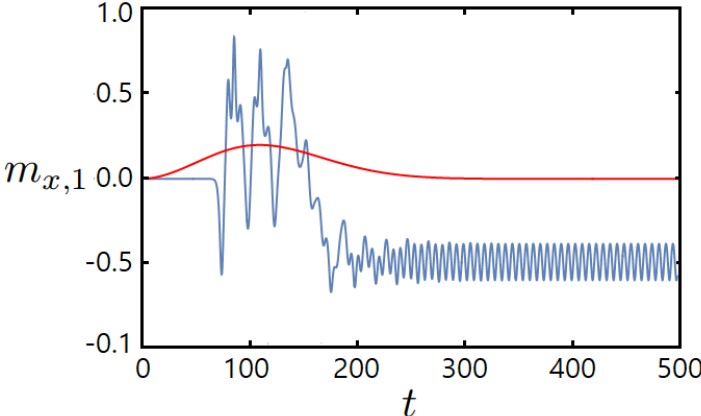

Figure 2: Time evolution of the order parameter of the low-energy sector (blue curve). The red curve corresponds to the envelope $E(t)$ of the perturbation, see Eq. (19), with $E_0 = 0.20$ and $\tau = 250$.

spins [23]. However, even in such case the model would remain simply a toy one, unable to describe any real solid-state material. For this reason, we prefer to maintain full connectivity, and introduce local dissipation via the Lindblad formalism. Similarly, we do not pretend to derive the Lindblad equations from any Hamiltonian of the system plus a bath, upon integrating out the latter. Instead, we here consider the most general Lindblad equation compatible with the mean-field character of the Hamiltonian (1) [1], and able to drive the system towards thermal equilibrium [24, 25]; specifically, compare with Eq. (21),

$$
\begin{aligned}
\frac{\partial \hat{\rho}_i(t)}{\partial t} = &-i\left[\hat{H}_i(t), \hat{\rho}_i(t)\right] \\
&+ \sum_{n<m}\Bigg[\gamma_{n\leftarrow m}(t)\Big(2\hat{L}_{n\leftarrow m}(t)\hat{\rho}_i(t)\hat{L}_{n\rightarrow m}(t) - \big\{\hat{L}_{n\rightarrow m}(t)\hat{L}_{n\leftarrow m}(t), \hat{\rho}_i(t)\big\}\Big) \\
&+ \gamma_{n\rightarrow m}(t)\Big(2\hat{L}_{n\rightarrow m}(t)\hat{\rho}_i(t)\hat{L}_{n\leftarrow m}(t) - \big\{\hat{L}_{n\leftarrow m}(t)\hat{L}_{n\rightarrow m}(t), \hat{\rho}_i(t)\big\}\Big)\Bigg],
\end{aligned}
\tag{24}
$$

where $\hat{H}_i(t)$ is still defined through Eqs. (22) and (23), while the Lindblad downward jump operators are

$$
\hat{L}_{n\leftarrow m}(t) \equiv |n; i, t\rangle \langle m; i, t|, \quad n < m,
\tag{25}
$$

where $|n; i, t\rangle$, $n = 0, \dots, 3$, is the instantaneous eigenstate of $\hat{H}_i(t)$ with eigenvalue $E_n(t)$, such that $E_0(t) \leq E_1(t) \leq E_2(t) \leq E_3(t)$ (this choice is called "self-consistent bath" and ensures that the system evolves towards its thermal equilibrium [24, 25]). Assuming $n < m$, we distinguish between downward jump operators, $\hat{L}_{n\leftarrow m}(t)$, see panel b) of Fig. 3, which correspond to de-excitations from a state to a lower energy one, and the reverse upward ones, $\hat{L}^\dagger_{n\leftarrow m}(t) \equiv L_{n\rightarrow m}(t)$. Lindblad operators can be easily expressed in terms of hard core bosons

---

[1] After resorting to the mean field approximation (let us remember that in our case it is exact in the thermodynamic limit), the Hamiltonian becomes a function of the density matrix. Indeed, the Hamiltonian parameters (the magnetic field in our case) should be determined by self-consistency conditions like in Eq.(23). It follows that the Hamiltonian eigenvalues become a function of the density matrix too. Now, as the set of Lindblad operators is written in terms of the Hamiltonian eigenstates, when a mean field approximation is performed the Lindblad equation becomes a non-linear differential equation with Lindblad operators that are implicitly a function of the density matrix. Furthermore, in principle a Lindblad operator could connect any two states within the Hilbert space of an interacting Hamiltonian but, after the mean field approximation, as the density matrix factorises into the product of single-site density matrices, the Lindblad operators only involves eigenstates at the same site. See Refs. [25, 38, 39] for some examples.

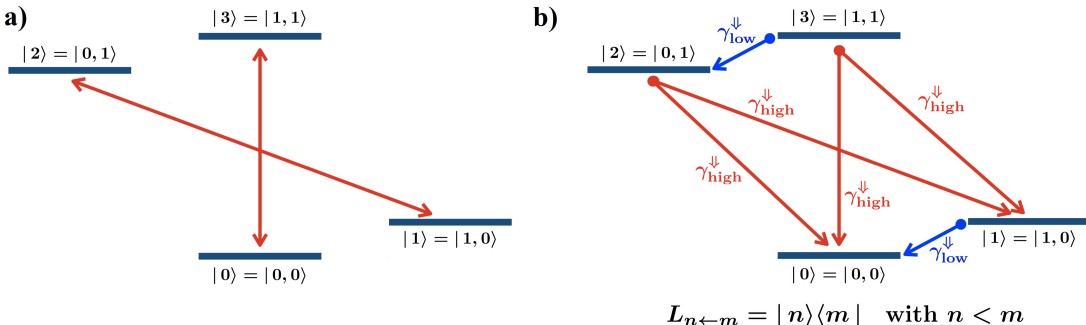

$$L_{n \leftarrow m} = |n\rangle\langle m| \quad \text{with } n < m$$

Figure 3: a) Transitions between the energy levels of the unperturbed system introduced through the perturbation Hamiltonian in Eq. (18) ; b) Downward jump operators, $L_{n \leftarrow m} = |n\rangle\langle m|$ with $n < m$, which, together with their hermitean conjugates, $L_{n \leftarrow m}^{\dagger} \equiv L_{n \rightarrow m}$, the upward jump operators, define the dissipative Lindblad dynamics. We distinguish high energy jump operators, in red, from low-energy ones, in blue. The former have coupling strength $\gamma_{\text{high}}^{\Downarrow}$, while the latter $\gamma_{\text{low}}^{\Downarrow}$.

through Eq. (16). Detailed balance, which ensures that the Boltzmann distribution is the stationary solution of Eq. (24), implies that

$$\gamma_{n \rightarrow m}(t) = e^{-\beta\left(E_m(t) - E_n(t)\right)} \gamma_{n \leftarrow m}(t), \tag{26}$$

where, since $n < m$, then $E_m(t) - E_n(t) > 0$, and therefore $\gamma_{n \rightarrow m}(t) < \gamma_{n \leftarrow m}(t)$. It follows that the Lindblad dynamics can be parametrised only through the six coupling strengths of the downward jump operators, $\gamma_{n \leftarrow m}(t)$ with $n < m$. In order to simplify the analysis, we assume that $\gamma_{n \leftarrow m}(t) = \gamma_{\text{high}}^{\Downarrow}$ for all the high-energy de-excitation processes $(n, m) = (0, 2), (0, 3), (1, 2), (1, 3)$, red arrows in panel b) of Fig. 3, distinct from $\gamma_{n \leftarrow m}(t) = \gamma_{\text{low}}^{\Downarrow}$ for the low-energy ones $(n, m) = (0, 1), (2, 3)$, blue arrows in panel b) of Fig. 3.

Since we expect that high-energy excitations dissipate faster than low-energy ones, we assume

$$\frac{\gamma_{\text{high}}^{\Downarrow}}{\gamma_{\text{low}}^{\Downarrow}} = r \geq 1. \tag{27}$$

Moreover, being the Lindblad equation (24) valid when the coupling to the dissipative bath is weak, we further take $\gamma_{\text{high}}^{\Downarrow} = 0.05$ small, so that all other coupling strengths, $\gamma_{\text{low}}^{\Downarrow}$ and the upward ones, see Eq. (26), are even smaller.

## 3.1 Time evolution upon changing bath and pulse parameters

In Fig. 4 we show the results of the numerical integration of Eq. (24) at temperature $T = 1.5\, T_c$, and for increasing $r$, see Eq. (27), from $r = 1$, blue line, to $r = 640$, purple line. The laser pulse parameters, see Eq. (19), are $E_0 = 0.2$ and $\tau = 1000$. As long as the laser pulse is on, the order parameter time evolution is similar to the non dissipative case shown in Fig. 2, i.e. it starts from $m_{x,1} = 0$, corresponding to the disordered system, and evolves toward one of the two $Z_2$-equivalent symmetry ordered phases. At the end of the laser pulse, while in the non dissipative case the system remains trapped in the ordered phase with $m_{x,1} \neq 0$, in the dissipative case the systems tends to return in the disordered one after a time that depends on the ratio $r$. We note that when low-energy excitations dissipate as fast as high-energy ones, blue line at $r = 1$, the system quickly relax to thermal equilibrium, $m_{x,1} = 0$, without showing

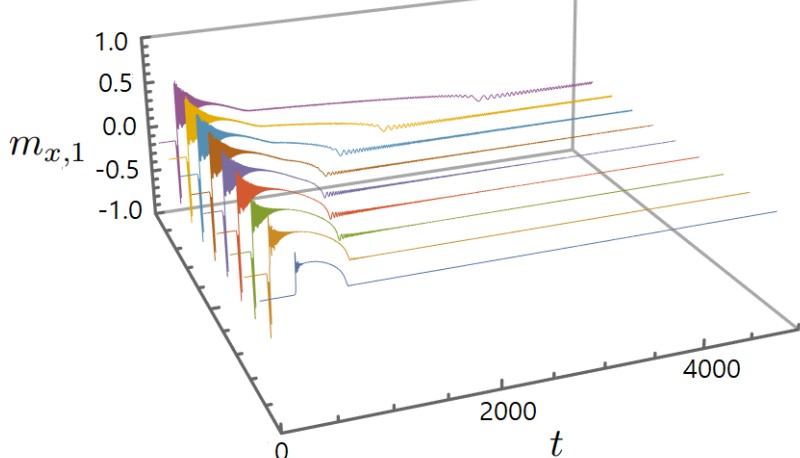

Figure 4: Time evolution of the order parameter $m_{x,1}$, compare with Fig. 2, for $r$ in Eq. (27) sets to $r = 1$ (blue curve), $r = 5$ (orange curve), $r = 10$ (green curve), $r = 20$ (red curve), $r = 40$ (blue curve), $r = 80$ (brown curve), $r = 160$ (light-blue curve), $r = 320$ (yellow curve) and, finally, $r = 640$ (purple curve). The pulse parameters are $E_0 = 0.2$ and $\tau = 1000$, see Eq. (19).

any transient cooling. The latter appears only upon increasing $r$, and lasts longer the larger $r$ is.

To quantify how long the system remains trapped into a non-thermal symmetry-broken state, we define a 'critical time' $t_{c,m}$ as the interval between the peak of the pulse, i.e., $t_{max} = \tau\sqrt{E_0}$, and the the time at which the magnetisation reaches its thermal value $m_{x,1} = 0$. Fig. 5 shows $t_{c,m}$ as a function of $r$, at fixed $E_0 = 0.14$ but different pulse durations $\tau$. We note that the critical time increases with $r$, as already highlighted in Fig. 4, but it is not monotonous with $\tau$. This may look counterintuitive, since one expects that a longer laser pulse at fixed $E_0$ transfers more energy from subsystem 1 to subsystem 2. However, a longer $\tau$ also allows energy to flow back in subsystem 1 before a quasi-stationary state is established, thus the non-monotonous behaviour. It follows that, if the perturbation lasts a time too short for dissipation to fully set in, its final effect critically depends whether, at the pulse end, the energy has reached a maximum

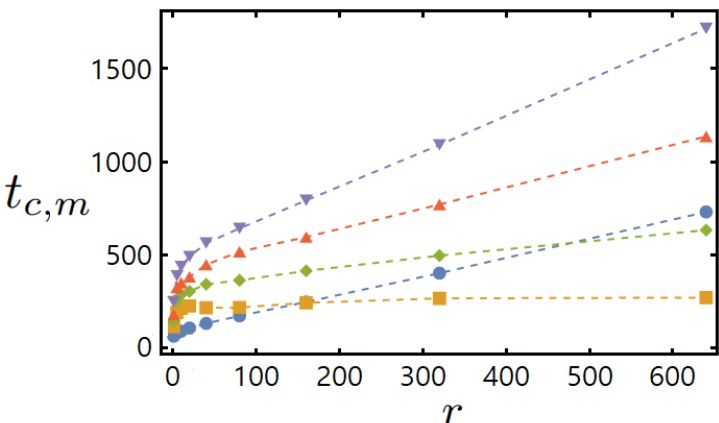

Figure 5: Critical time $t_{c,m}$ for $E_0 = 0.14$ and: $\tau = 250$ (blue circles), $\tau = 500$ (orange squares), $\tau = 650$ (green diamonds), $\tau = 800$ (red up triangles), $\tau = 1000$ (purple down triangles).

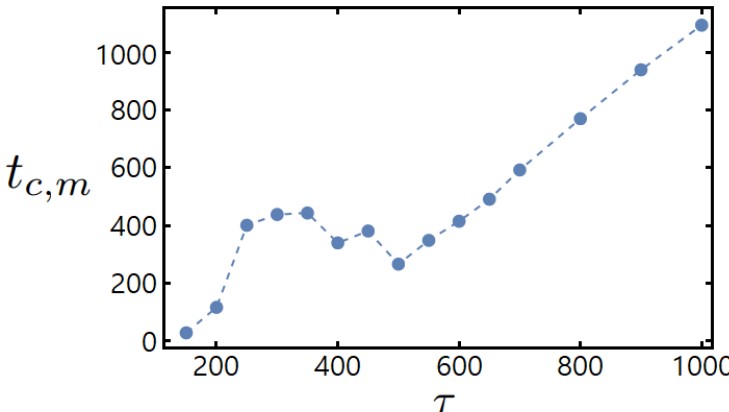

Figure 6: Critical time $t_{c,m}$ as a function of $\tau$, for $r = 320$ and $E_0 = 0.14$.

or a minimum of its evolution. On the contrary, if the pulse duration is longer than the typical timescale of dissipation, such memory effect is lost. To stress this behaviour, in Fig. 6 we plot $t_{c,m}$ as a function of $\tau$ at fixed $E_0 = 0.14$ and $r = 320$. We observe that for $\tau < 500$ the critical time is not monotonous, while it becomes so only for larger $\tau$, where it grows linearly with the pulse duration.

To complete our analysis of the 'critical time' dependence upon the bath and pulse parameters, in panel a) of Fig. 8 we plot $t_{c,m}$ as function of $E_0$, at fixed $r = 640$ and for three different values of $\tau$. In conclusion, when the pulse duration is long enough to make dissipation active well before the pulse end, the time $t_{c,m}$ during which the system is trapped into a non-thermal symmetry broken state increases monotonously with $r$, $\tau$ and $E_0$.
In Fig. 7 we show the behaviour of $t_{c,m}$ as a function of the bath temperature $T$, at fixed $r = 640$, $\tau = 750$ and $E_0 = 0.2$. We observe that the critical time diverges as the temperature approaches the critical value $T_c$, see Eq. (9), with a mean-field-like critical behaviour. Indeed, the numerical data are well approximated by the fitting function $f(T) \approx 970 \left( \frac{T - T_c}{T_c} \right)^{-1}$.
Finally, in panel b) of Fig. 8 we plot the value of the order parameter of subsystem 1, $m_{x,1}$, as a function of $E_0$, at fixed $r = 640$ and $\tau = 1200$ and after a time $t = 2000$ measured from the peak of the pulse. The order parameter exhibits a monotonous increase with $E_0$, that is

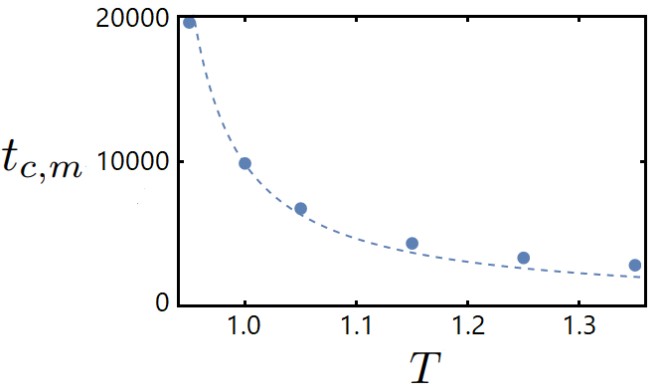

Figure 7: Critical time $t_{c,m}$ as a function of the temperature $T$, for $r = 640$, $E_0 = 0.2$ and $\tau = 750$. The dashed line is the fitting function $f(T) \approx 970 \left( \frac{T - T_c}{T_c} \right)^{-1}$, with $T_c$ given by Eq. (9).

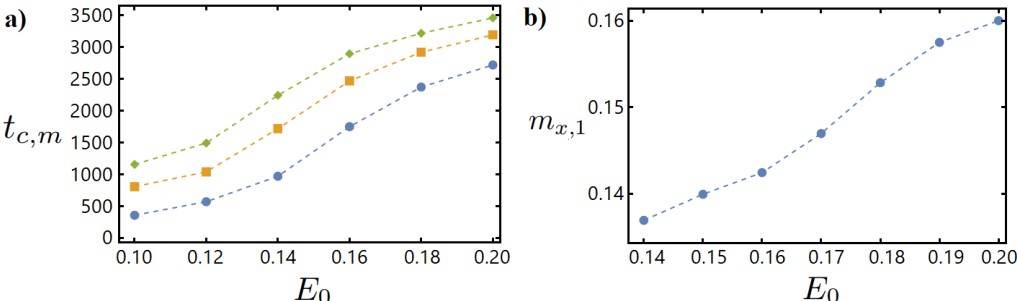

Figure 8: a) Critical time $t_{c,m}$ as a function of $E_0$, for $r = 640$ and: $\tau = 750$ (blue curve), $\tau = 1000$ (orange curve), $\tau = 1200$ (green curve); b) order parameter $m_{x,1}$ as a function of $E_0$, for $r = 640$ and $\tau = 1200$ at $t = 2000$ (measured from the peak of the pulse $t_{max}$).

with the energy irradiated by the laser pulse, in agreement with the recent experiments in $K_3C_{60}$ [14]. Indeed, panel b) of Fig. 8 should be qualitatively compared with panel b) of Fig.5 of Ref. [14] where it emerge that a stronger laser pulse, at fixed pulse length and pump-pulse duration, drives the system deeper in the metastable phase.

## 3.2 Time evolution at constant pulse 'fluence'

Until now, we have compared results obtained for perturbations having the same amplitude $E_0$ or the same duration $\tau$. In what follows, we study the system response upon increasing the laser pulse duration $\tau$ while properly reducing its peak amplitude so as to maintain constant the total supplied energy, defined as [26]

$$F = \int_0^\infty |E(t)|^2 \, dt \,, \tag{28}$$

which can be regarded as the pulse 'fluence'. In Figs. 9 and 10 we show the time evolution of the order parameter, $m_{x,1}$, for increasing $\tau$ at constant $F = 15.53$, thus decreasing $E_0$ correspondingly.

We observe that, while for short times $m_{x,1}$ peaks more in presence of a spiked pulse rather than a longer but flatter one, for long times the latter is much more efficient to make a finite $m_{x,1}$ survive longer. Panel a) of Fig. 11 shows that the critical time, $t_{c,m}$, indeed grows substantially with $\tau$ at fixed $F$. Essentially, for large $\tau$, the transient non-thermal symmetry-broken phase becomes a quasi-steady state kept alive by the dissipative bath. In order to compare the qualitative behaviour of the time evolution at constant pulse 'fluence' of our toy model with the experimental results of Ref. [14], in panel b) of Fig. 11 we plot the order parameter of subsystem 1, $m_{x,1}$, measured right after the end of the laser pulse, as a function of $\tau$. We observe a quite flat dependence of the order parameter by the laser pulse length if the laser pulse 'fluence' is kept constant. Again, the result shows an interesting qualitative agreement with the experimental observation reported in panel a) of Fig.5 of Ref. [14] where, indeed, 'the photoresistivity was mostly independent of the pump-pulse duration and depended only on the total energy of the excitation pulse'. Indeed, as observed in Ref. [14], this is a rather interesting aspect as the fact that the amplitude of the effect only depends on the laser pulse fluence is a strong evidence against the early proposed approaches based on the non-linear phonon mechanism where the system response is expected to depend on the laser pulse peak amplitude [12] while it looks to be consistent with the selective cooling mechanism.

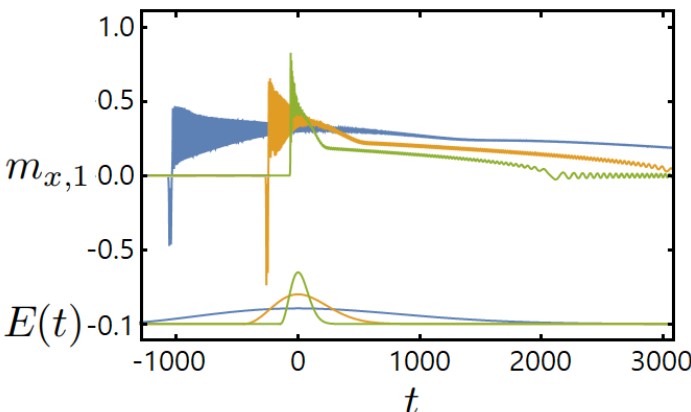

Figure 9: Time evolution of the order parameter, $m_{x,1}$, at $r = 640$ and different values of $\tau$ and $E_0$ such that $F$ in Eq. (28) is kept constant at the value 15.53. In particular, we use $\tau = 5000$ and $E_0 = 0.105$ (blue curve), $\tau = 1000$ and $E_0 = 0.2$ (orange curve), $\tau = 250$ and $E_0 = 0.35$ (green curve). The curves have been shifted so that $t = 0$ corresponds to the peak amplitude of the pulses, shown in the lower part of the plot.

## 4 Conclusions

In this paper we have investigated the transient cooling mechanism brought on by a pulse perturbation in the two coupled infinite-range quantum Ising models of Ref. [21], but now in presence of dissipation. We have shown that the cooling of the low-energy degrees of freedom at the expense of the high-energy ones, observed in absence of dissipation, is not spoiled in its presence, especially when excitations dissipate faster the higher their energy. On the contrary, dissipation enhances the cooling effect of the perturbation, stabilising a non-thermal quasi-steady state that lasts for long after the pulse end. Specifically, we have found that increasing the pulse duration keeping the 'fluence', $F$ of Eq. (28), constant, makes such quasi-steady state survive longer and longer, not in disagreement with recent experiments in $K_3C_{60}$ [14]. It is worth stressing that, while we make some not unphysical assumptions on the dissipative processes, we also show that the selective cooling persists for a wide range of parameters. Indeed, we just assume that high-energy excitations dissipate faster than low-energy ones, which is a rather generic circumstance. In reality, the most critical parameter here is the strength of the coupling to the 'laser pulse', i.e., the dipole matrix element in real materials, or, equivalently, the intensity of the optical absorption that is related to the strength of the transition processes. In our toy model there is just a single transition process that connects the low energy sector, states 0 and 1, with the high energy one, states 2 and 3. Evidently, the cooling is less efficient the smaller the coupling strength. However, regarding $K_3C_{60}$, we recall that in the experiment the laser hits a rather pronounced mid-infrared peak, which we have associated to an exciton that plays the role of the high-energy sector [20], while the role of the low energy sector is there played by particle-hole excitations. The intensity of that peak reflects the substantial strength of the process. It is self-evident that it would be totally useless to pump at an excitation with very small absorption strength. Indeed, the strength of the optical process is a prerequisite of our cooling strategy.

Evidently, our toy model is an extreme oversimplification of any real material. However, the key ingredients that make the cooling strategy successful may characterise many real systems, especially strongly correlated ones where localised atomic-like high-energy excitations coexist with low-energy coherent particle-hole excitations, like, e.g., in $K_3C_{60}$.

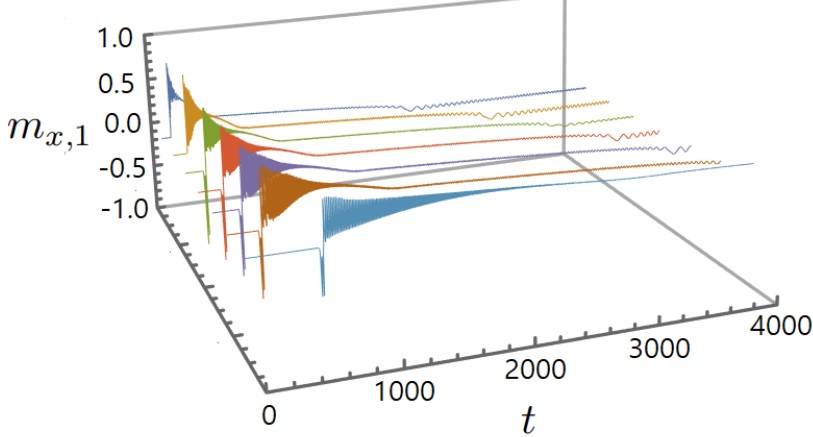

Figure 10: Same as in Fig.9 without the time shift, and for $\tau = 250$ (blue curve), $\tau = 500$ (orange curve), $\tau = 750$ (green curve), $\tau = 1000$ (red curve), $\tau = 1250$ (purple curve), $\tau = 1500$ (brown curve), $\tau = 5000$ (light blue curve).

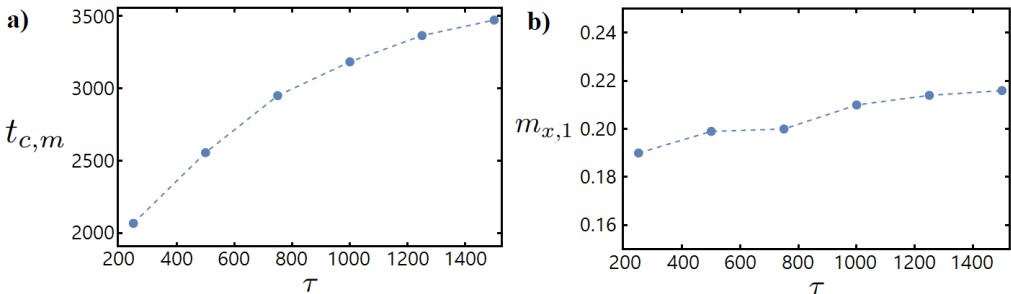

Figure 11: a) Critical time $t_{c,m}$ as a function of $\tau$. The amplitude $E_0(\tau)$ is chosen so as to maintain constant the 'fluence' in Eq. (28); b) order parameter $m_{x,1}$ as a function of $\tau$, for $r = 640$ and constant 'fluence', measured at time $t$ right after the end of the laser pulse.

Such dissipative cooling effect resembles much the nuclear Overhauser effect [27–29], which has been also realised by optical pumping [30], and, to some extent, laser cooling in optomechanical systems, where mechanical oscillators can be cooled by coupling them to a microwave cavity [31–33]. However, in our case the role of entropy source and sink are played by the low- and high-energy internal degree of freedom of the system, without the need of an optical or microwave cavity. As a consequence, in our model the transient cooling does not suffer from the stringent limitations given, e.g., by the microwave cavity decay rate, which strongly affect optomechanical cooling, and make challenging its experimental realisation [34].

We finally mention recent works [35–37] that show how properly designed dissipative protocols can efficiently prepare a quantum system in its ground state. Therefore, tailoring dissipation may represent a novel tool to control open many-body systems, including the selective cooling that we have here discussed.

**Funding information** A. N. was financially supported by POR Calabria FESR-FSE 2014/2020 - Linea B) Azione 10.5.12, grant no. A.5.1. M. F. acknowledges financial support from the European Research Council (ERC) under the European Union's Horizon 2020 research and innovation programme, Grant agreement No. 692670 "FIRSTORM".

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
