# Peer review of "Dissipative cooling induced by pulse perturbations"

_SciPost Physics, doi:SciPost Phys. 12, 014 (2022)_

## Round 2 · Referee Report · Anonymous (Referee 3) · 2021-9-27

Report

I feel that the authors have not taken into account all the comments in the referee reports.
For example the question of referee two why the laser-interaction appears as an interaction between two spins is explained in the response of the referees but not in the manuscript.
I agree with the referee that this aspect is not clear in the manuscript, and I do feel that it should be clarified there.

I also feel that the discussion of the explicit results has hardly improved. Fig. 4 for example shows several complicated functions, but there is no explanation of what features in the functions are relevant. The authors essentially assume that the reader will understand the results on their own.
Later-on, the authors show Fig. 10 with the only comments that it is similar data to what is shown in Fig. 4.
I don't think that many readers will be able to appreciate the results.

While I still think that the actual results might warrant publication, I feel that the explanations are substantially too scarce to make the manuscript reasonably accessible.
I thus maintain my assessment that publication can be recommended only after a substantial revision.

---

## Round 2 · Referee Report · Anonymous (Referee 2) · 2021-10-12

Report

[Q1] I wonder why the authors are introducing the hard core bosons in Eq. (15). They do not seem to be used later on, and the discussion concerning the structure of the spectrum [Eqs. (14)] could also be performed without them. Is there any profound reason for the introduction of hard core bosons?

[A1] The hard core bosons are introduced to make, at least in our opinion, more transparent the definition of the four eigenstates in Eq.[16] and the role of the pulse perturbation. For that reason, in the revised version we have written the pulse perturbation Eq.[18] explicitly in terms of the hard core bosons, which also answers the next request (see [R2]-[A2]).

[Response1] I still don't think that introducing the notion "hard core bosons" is very helpful, but I leave the choice with the authors. I would just ask the authors to please write a few more details or at least mention that this is merely a formal construction. Just introducing the bosons by writing "In other words, we can write..." is not helpful. Also, please use the bosons consistently; e.g. below Eq. (19) you resort to projectors/ket-bra symbols, that could all be represented by bosonic operators for consistency. The same for the jump operators; those could also be expressed in terms of hard core bosons, right?

[Q2] I do not quite understand why Eq. (18) is mimicking a laser pulse. In my opinion a laser acts at the level of a single particle (due to the dipolar selection rule). Therefore the corresponding operator should be a sum of single body operators. Where does the \sigma_x \sigma_x - term come from? This rather resembles a (dipolar) interaction.

[A2] In our model spins and, equivalently, hard core bosons are just a tool to represent a Hilbert space comprising four energy states on each site, two, 0 and 1, with lower energy and two, 2 and 3, with higher one. The laserps–e allows transitions between the two sectors, specifically, 0 ↔︎ 3 and 1 ↔︎ 2, which are simply interpreted in terms of hard-core bosons. The key to our mechanism is to tune the laser frequency in resonance with 1 ↔︎ 2, thus depleting the occupation of boson b1 and increasing that of b2. The optical analogy can be make more evident imagining that b1 is even under parity and b2 odd, so that all transition processes 0 ↔︎ 3 and 1 ↔︎ 2 are dipole active.

[Pag 6] Modified Eq.[18] to express it in terms of the hard core bosons and added “The perturbation (18) allows for the transition processes b†2,ib\dagga1,i=\ket2;i\bra1;i and b†2,ib†1,i=\ket3;i\bra0;i plus their hermitean conjugates (as depicted in panel a) of Fig.~???). The optical analogy becomes evident if we assume b\dagga1,i even under parity and b\dagga2,i odd, so that all the above transition processes couple opposite parity states.”

[Response2] Thanks for the clarification. Please rework the sentence that you added to the manuscript. It is not clear what you mean by "optical analogy" (the reader does not know about our exchange here). This is just another example (together with the hard core bosons), which shows that the manuscript is too cryptic and implicit in many place. Please spell out, what you want to say.

[Q3] The notation of Eq. (23) is confusing when compared with Eq. (8). In one expression the argument is time while in the other one it is temperature.

[A3] In Eq.[8] the order parameter and the density matrix are the ones at equilibrium at temperature T, in Eq.[23] the system is out-of-equilibrium and the density matrix becomes time-dependent and reach the equilibrium one at t→∞. In order to distinguish the equilibrium and out-of-equilibrium cases we added the suffix (eq) in the formulas of Sec.(2.1).

[Response3] Thanks.

[Q4] I find the motivation of the dissipative terms not very convincing. Is it justified to assume that the rates are explicitly time-dependent, i.e. isn't there some implicit separation of time scales that is assumed to hold?

[A4] We mention that all the results do not change appreciably if we use the rate corresponding to the equilibrium state and represent the instantaneous eigenstates in terms of the equilibrium ones. However, we decided to use time dependent rates to be consistent with the mean field approximation and the detailed balance condition required to reach equilibrium (see Ref.[25] for a comparison between self-consistent and fixed baths in a simple case, and Ref.[24] for a general discussion).

[Page 7] “Instead, we here consider the most general Lindblad equation compatible with the mean-field character of the Hamiltonian (1), and able to drive the system towards thermal equilibrium \cite[24,25]. […] (this choice is called "self-consistent bath" and ensures that the system evolves towards its thermal equilibrium \cite{24,25}).”

Where: [24]: N. Lang and H. P. Büchler, Rev. A92, 012128 (2015), Exploring quantum phases by driven dissipation; [25]: D. S. Kosov, T. Prosen and B. Zunkovic, Journal of Physics A: Mathematical and Theoretical Phys. 44(46), 462001 (2011), Lindblad master equation approach to superconductivity in open quantum systems.

[Response4] Can you please spell out what "compatibility with the mean-field character" precisely means?

[Q5] In the conclusions you write "On the contrary, dissipation enhances the cooling effect of the perturbation, stabilising a non-thermal quasi-steady state that lasts for long after the pulse end." Are the authors sure that the dissipation is not just constructed in a way that this is the case. There are a number of assumptions, e.g. concerning the ratio of the rates, which are to some extend arbitrary. Notes, that the relative energy difference is only one of the quantities that is entering the transition rates. There is also a kinetic part which determines whether a transition can take place (states with large energy difference may be connected by complicated transition paths). Especially for correlated systems this seems to be a relevant aspect. In this sense I find the connection to photo-induced superconductivity in K3C60 also a bit far-fetched.

[A5] The cooling mechanism does not require fine tuning and persists for a wide range of parameters. We just assume that high-energy excitations dissipate faster than low-energy ones, which is a rather generic circumstance. Another issue is the one related to the strength of the transition processes. In our toy model there is just a single one that connects the low energy sector, states 0 and 1, with the high energy one, states 2 and 3. Evidently, the cooling is less efficient the smaller the coupling strength. However, regarding K3C60, we recall the Referee that in the experiment the laser hits a rather pronounced mid-infrared peak, which we have associated to an exciton that plays the role of the high-energy sector, while the role of the low energy sector is there played by particle-hole excitations. The intensity of that peak reflects the substantial strength of the process. It is self-evident that it would be totally useless to pump at an excitation with very small absorption strength. Indeed, the strength of the optical process is a prerequisite of our cooling strategy, as explicitly mentioned in the introduction when we write “… essentially based on the existence of a high energy localised mode that, when the laser is on, is able to fast soak up entropy…”.

[Page 13] “It is worth stressing that, while we make some not unphysical assumptions on the dissipative processes, we also show that the selective cooling persists for a wide range of parameters. In reality, the most critical parameter here is the strength of the coupling to the `laser pulse', i.e., the dipole matrix element in real materials, or, equivalently, the intensity of optical absorption.”

[Response5] Thank you, I accept that. But could you please explain in what sense the strength of the coupling to the laser pulse is the "most critical parameter"? Also, rather than writing "we make some not unphysical assumptions", you could just spell out once more what assumptions you are making. And after that you can re-iterate why they are reasonable/physical.

---

## Round 2 · Referee Report · Anonymous (Referee 1) · 2021-10-13

Report

I think the resubmission satisfactorily addresses all previous concerns and the manuscript could be published in SciPost Physics. The expression of the interaction with the laser field in terms of two distinct spins actually makes sense because of the vastly different energy scales being involved. Making this point more explicit in the manuscript might be a good idea, but it is not strictly necessary for the understanding of the paper. I also think that the authors comment on Fig. 4 in the main text in an adequate way, especially concerning the influence of $r$ on relaxation time.

---

## Round 2 · Author Response

Dear Editor-in-charge,

We have considered the Referee reports concerning our manuscript, “Dissipative cooling induced by pulse perturbations”, submitted for publication to SciPost.

We thank the Referees for their constructive remarks, which we address by pertinently updating the revised version of our manuscript, as we detail in the following. Below, we also provide a concise answer to all the Referees' remarks, as well as a list of the major changes we performed to our paper.

We believe that, after the improvements we have made to our manuscript in addressing the Referees' remarks, our paper now meets the requirements to warrant publication in SciPost and resubmit it accordingly.

We thank you for your kind attention in our submission.

Best Regards

Andrea Nava (on behalf of all the authors)

Answers to: Anonymous Report 1 on 2021-7-2 (Invited Report)

We thank the Referee for her/his kind appreciation of our work.

[Q1] The idea to use driven-dissipative dynamics to cool strongly interacting many-body systems has also been investigated recently in the context of reservoir engineering techniques (e.g., [New J. Phys. 15, 073027; Science Adv. 6, eaaw9268; Phys. Rev. Research 2, 023214]). The authors should discuss how their approach is related these works.

[A1] We thank the Referee for pointing to our attention those references that provide very interesting contributions to thermalization and cooling of open quantum systems. We now cite them as Refs. [35-37] in the concluding section of our manuscript. Specifically:

[Page 14] “We finally mention recent works [35-37] that show how properly designed dissipative protocols can efficiently prepare a quantum system in its ground state. Therefore, tailoring dissipation may represent a novel tool to control open many-body systems, including the selective cooling that we have here discussed.”

[Q2] What is the total area of the pulse in the parameterization chosen by the authors? Is it possible for the energy to flow back to the system 1 or do the authors consider the equivalent of a \pi pulse between the two systems?

[A2] The laser pulse has a high oscillatory behaviour (see Eq.[18]), with envelope given by Eq.[19] and period T=2π/ω much lower of the envelope decay time τ. The total area of the pulse is therefore approximately vanishing, while the total supplied energy is given by Eq.[28]. During the laser pulse duration the energy can flow in both directions, (i.e. back and forth between subsystem 1 and subsystem 2), thus exhibiting an oscillatory behaviour (see Figs.[5,6] of Ref.[21]: Phys. Rev. Lett. 120, 220601 (2018)) but with the overall effect to cool down subsystem 1.

[Page 9] “However, a longer τ also allows energy to flow back in subsystem 1 before a quasi-stationary state is established, thus the non-monotonous behaviour.”

[Q3] Is the critical time somehow related to the critical properties of the underlying phase transition of the Ising model?

[A3] We would like to thank the referee for her/his interesting comment. In order to address this point, in the revised version of the manuscript we added Fig.[7] where we show the behaviour of the critical time as a function of the temperature while we approach the critical one. It emerges that the critical time diverges near $T_c$ with a mean-field-like critical behaviour.

[Page 10] “In Fig.~\ref{fig:critical} we show the behaviour of $t_{c,m}$ as a function of the bath temperature $T$, at fixed $r=640$, $\tau=750$ and $E_0=0.2$. We observe that the critical time diverges as the temperature approaches the critical value $T_c$, see Eq.~\eqn{critT}, with a mean-field-like critical behaviour. Indeed, the numerical data are well approximated by the fitting function $f\left(T\right)\approx 970 \left(\frac{T-T_c}{T_c}\right)^{-1}$.”

Answers to: Anonymous Report 2 on 2021-7-2 (Contributed Report)

We thank the Referee for her/his observations.

[Q1] I wonder why the authors are introducing the hard core bosons in Eq. (15). They do not seem to be used later on, and the discussion concerning the structure of the spectrum [Eqs. (14)] could also be performed without them. Is there any profound reason for the introduction of hard core bosons?

[A1] The hard core bosons are introduced to make, at least in our opinion, more transparent the definition of the four eigenstates in Eq.[16] and the role of the pulse perturbation. For that reason, in the revised version we have written the pulse perturbation Eq.[18] explicitly in terms of the hard core bosons, which also answers the next request (see [R2]-[A2]).

[Q2] I do not quite understand why Eq. (18) is mimicking a laser pulse. In my opinion a laser acts at the level of a single particle (due to the dipolar selection rule). Therefore the corresponding operator should be a sum of single body operators. Where does the \sigma_x \sigma_x - term come from? This rather resembles a (dipolar) interaction.

[A2] In our model spins and, equivalently, hard core bosons are just a tool to represent a Hilbert space comprising four energy states on each site, two, 0 and 1, with lower energy and two, 2 and 3, with higher one. The 'laser pulse' allows transitions between the two sectors, specifically, 0 ↔︎ 3 and 1 ↔︎ 2, which are simply interpreted in terms of hard-core bosons. The key to our mechanism is to tune the laser frequency in resonance with 1 ↔︎ 2, thus depleting the occupation of boson b1 and increasing that of b2. The optical analogy can be make more evident imagining that b1 is even under parity and b2 odd, so that all transition processes 0 ↔︎ 3 and 1 ↔︎ 2 are dipole active.

[Pag 6] Modified Eq.[18] to express it in terms of the hard core bosons and added “The perturbation (18) allows for the transition processes $b^\dagger_{2,i}\,b^\dagga_{1,i} = \ket{2;i}\bra{1;i}$ and $b^\dagger_{2,i}\,b^\dagger_{1,i} = \ket{3;i}\bra{0;i}$ plus their hermitean conjugates (as depicted in panel a) of Fig.~\ref{fig:levels}). The optical analogy becomes evident if we assume $b^\dagga_{1,i}$ even under parity and $b^\dagga_{2,i}$ odd, so that all the above transition processes couple opposite parity states.”

[Q3] The notation of Eq. (23) is confusing when compared with Eq. (8). In one expression the argument is time while in the other one it is temperature.

[A3] In Eq.[8] the order parameter and the density matrix are the ones at equilibrium at temperature T, in Eq.[23] the system is out-of-equilibrium and the density matrix becomes time-dependent and reach the equilibrium one at $t \rightarrow \infty$. In order to distinguish the equilibrium and out-of-equilibrium cases we added the suffix $(eq)$ in the formulas of Sec.(2.1).

[Q4] I find the motivation of the dissipative terms not very convincing. Is it justified to assume that the rates are explicitly time-dependent, i.e. isn't there some implicit separation of time scales that is assumed to hold?

[A4] We mention that all the results do not change appreciably if we use the rate corresponding to the equilibrium state and represent the instantaneous eigenstates in terms of the equilibrium ones. However, we decided to use time dependent rates to be consistent with the mean field approximation and the detailed balance condition required to reach equilibrium (see Ref.[25] for a comparison between self-consistent and fixed baths in a simple case, and Ref.[24] for a general discussion).

[Page 7] “Instead, we here consider the most general Lindblad equation compatible with the mean-field character of the Hamiltonian (1), and able to drive the system towards thermal equilibrium \cite[24,25]. […] (this choice is called "self-consistent bath" and ensures that the system evolves towards its thermal equilibrium \cite{24,25}).”

Where: [24]: N. Lang and H. P. Büchler, Rev. A92, 012128 (2015), Exploring quantum phases by driven dissipation; [25]: D. S. Kosov, T. Prosen and B. Zunkovic, Journal of Physics A: Mathematical and Theoretical Phys. 44(46), 462001 (2011), Lindblad master equation approach to superconductivity in open quantum systems.

[Q5] In the conclusions you write "On the contrary, dissipation enhances the cooling effect of the perturbation, stabilising a non-thermal quasi-steady state that lasts for long after the pulse end." Are the authors sure that the dissipation is not just constructed in a way that this is the case. There are a number of assumptions, e.g. concerning the ratio of the rates, which are to some extend arbitrary. Notes, that the relative energy difference is only one of the quantities that is entering the transition rates. There is also a kinetic part which determines whether a transition can take place (states with large energy difference may be connected by complicated transition paths). Especially for correlated systems this seems to be a relevant aspect. In this sense I find the connection to photo-induced superconductivity in K3C60 also a bit far-fetched.

[A5] The cooling mechanism does not require fine tuning and persists for a wide range of parameters. We just assume that high-energy excitations dissipate faster than low-energy ones, which is a rather generic circumstance. Another issue is the one related to the strength of the transition processes. In our toy model there is just a single one that connects the low energy sector, states 0 and 1, with the high energy one, states 2 and 3. Evidently, the cooling is less efficient the smaller the coupling strength. However, regarding K3C60, we recall the Referee that in the experiment the laser hits a rather pronounced mid-infrared peak, which we have associated to an exciton that plays the role of the high-energy sector, while the role of the low energy sector is there played by particle-hole excitations. The intensity of that peak reflects the substantial strength of the process. It is self-evident that it would be totally useless to pump at an excitation with very small absorption strength. Indeed, the strength of the optical process is a prerequisite of our cooling strategy, as explicitly mentioned in the introduction when we write “… essentially based on the existence of a high energy localised mode that, when the laser is on, is able to fast soak up entropy…”.

[Page 13] “It is worth stressing that, while we make some not unphysical assumptions on the dissipative processes, we also show that the selective cooling persists for a wide range of parameters. In reality, the most critical parameter here is the strength of the coupling to the `laser pulse', i.e., the dipole matrix element in real materials, or, equivalently, the intensity of optical absorption.”

Answers to: Anonymous Report 3 on 2021-7-5 (Invited Report)

We thank the Referee for her/his kind appreciation of our work.

[Q1] I am not sure how realistic the considered model of two coupled Ising Hamiltonians is; maybe the authors can comment on that in a revised manuscript.

[A1] While the toy model we propose is extremely simple, it contains the basic ingredients to describe the generic selective cooling mechanism that takes place in more realistic systems like, we believe, alkali doped fullerides.

[Page 13] “Evidently, our toy model is an extreme oversimplification of any real material. However, the key ingredients that make the cooling strategy successful may characterise many real systems, especially strongly correlated ones where localised atomic-like high-energy excitations coexist with low-energy coherent particle-hole excitations, like, e.g., in K$_3$C$_{60}$.”

[Q2] I feel that the discussion in the manuscript is a bit brief. […]

[A2] We thank the referee for her/his comment, In the updated version of the manuscript we extended the discussion in Sec.3.1, Sec.3.2 and Sec.4 and modified Fig.[8] and Fig.[11] to make a direct comparison with the recent experimental results on K3C60 of Ref.[14]. We also added Fig.[7] to discuss the behaviour of the order parameter with the temperature. Moreover, we tried to make the presentation less colloquial.

---

## Round 2 · List of Changes

Warnings issued while processing user-supplied markup:

  • Inconsistency: plain/Markdown and reStructuredText syntaxes are mixed. Markdown will be used.
    Add "#coerce:reST" or "#coerce:plain" as the first line of your text to force reStructuredText or no markup.
    You may also contact the helpdesk if the formatting is incorrect and you are unable to edit your text.

List of major changes

[Sec.2.1] we added the suffix $(eq)$ to the density matrix and the order parameter to distinguish them from the time dependent ones.

[Pag 6] Modified Eq.[18] to express it in terms of the hard core bosons and added “The perturbation (18) allows for the transition processes $b^\dagger_{2,i}\,b^\dagga_{1,i} = \ket{2;i}\bra{1;i}$ and $b^\dagger_{2,i}\,b^\dagger_{1,i} = \ket{3;i}\bra{0;i}$ plus their hermitean conjugates (as depicted in panel a) of Fig.~\ref{fig:levels}). The optical analogy becomes evident if we assume $b^\dagga_{1,i}$ even under parity and $b^\dagga_{2,i}$ odd, so that all the above transition processes couple opposite parity states.”

[Page 7] “Instead, we here consider the most general Lindblad equation compatible with the mean-field character of the Hamiltonian (1), and able to drive the system towards thermal equilibrium \cite[24,25]. […] (this choice is called "self-consistent bath" and ensures that the system evolves towards its thermal equilibrium \cite{24,25}).”

[Page 9] “However, a longer τ also allows energy to flow back in subsystem 1 before a quasi-stationary state is established, thus the non-monotonous behaviour.”

[Page 10] “In Fig.~\ref{fig:critical} we show the behaviour of $t_{c,m}$ as a function of the bath temperature $T$, at fixed $r=640$, $\tau=750$ and $E_0=0.2$. We observe that the critical time diverges as the temperature approaches the critical value $T_c$, see Eq.~\eqn{critT}, with a mean-field-like critical behaviour. Indeed, the numerical data are well approximated by the fitting function $f\left(T\right)\approx 970 \left(\frac{T-T_c}{T_c}\right)^{-1}$.”

[Page 11] “Finally, in panel b) of Fig.~\ref{fig:varE0} we plot the value of the order parameter of subsystem 1, $m_{x,1}$, as a function of $E_0$, at fixed $r=640$ and $\tau=1200$ and after a time $t=2000$ measured from the peak of the pulse. The order parameter exhibits a monotonous increase with $E_0$, that is with the energy irradiated by the laser pulse, in agreement with the recent experiments in K$_3$C$_{60}$~\cite{Cavalleri-K3C60-2020}. Indeed, panel b) of Fig.~\ref{fig:varE0} should be qualitatively compared with panel b) of Fig.5 of Ref.~\cite{Cavalleri-K3C60-2020} where it emerge that a stronger laser pulse, at fixed pulse length and pump-pulse duration, drives the system deeper in the metastable phase.”

[Page 12] “Essentially, for large $\tau$, the transient non-thermal symmetry-broken phase becomes a quasi-steady state kept alive by the dissipative bath. In order to compare the qualitative behaviour of the time evolution at constant pulse 'fluence' of our toy model with the experimental results of Ref.~\cite{Cavalleri-K3C60-2020}, in panel b) of Fig.~\ref{fig:tc-tmax_samearea} we plot the order parameter of subsystem 1, $m_{x,1}$, measured right after the end of the laser pulse, as a function of $\tau$. We observe a quite flat dependence of the order parameter by the laser pulse length if the laser pulse 'fluence' is kept constant. Again, the result shows an interesting qualitative agreement with the experimental observation reported in panel a) of Fig.5 of Ref.~\cite{Cavalleri-K3C60-2020} where, indeed, 'the photoresistivity was mostly independent of the pump-pulse duration and depended only on the total energy of the excitation pulse'.”

[Page 13] “It is worth stressing that, while we make some not unphysical assumptions on the dissipative processes, we also show that the selective cooling persists for a wide range of parameters. In reality, the most critical parameter here is the strength of the coupling to the `laser pulse', i.e., the dipole matrix element in real materials, or, equivalently, the intensity of optical absorption.”

[Page 13] “Evidently, our toy model is an extreme oversimplification of any real material. However, the key ingredients that make the cooling strategy successful may characterise many real systems, especially strongly correlated ones where localised atomic-like high-energy excitations coexist with low-energy coherent particle-hole excitations, like, e.g., in K$_3$C$_{60}$.”

[Page 14] “We finally mention recent works [35-37] that show how properly designed dissipative protocols can efficiently prepare a quantum system in its ground state. Therefore, tailoring dissipation may represent a novel tool to control open many-body systems, including the selective cooling that we have here discussed.”

We added one panel to Figs.[3,8,11], and modified the captions, in agreement with the changes in the text listed above. We added Fig.[7].

We added references: [24]: N. Lang and H. P. Büchler, Rev. A92, 012128 (2015), Exploring quantum phases by driven dissipation; [25]: D. S. Kosov, T. Prosen and B. Zunkovic, Journal of Physics A: Mathematical and Theoretical Phys. 44(46), 462001 (2011), Lindblad master equation approach to superconductivity in open quantum systems; [35]: C. Cormick, A. Bermudez, S. F. Huelga and M. B. Plenio, New Journal of Physics 15(7), 073027 (2013), Dissipative ground state preparation of a spin chain by a structured environment; [36]: M. Raghunandan, F. Wolf, C. Ospelkaus, P. O. Schmidt and H. Weimer, Science Advances 6(10) (2010), Initialization of quantum simulators by sympathetic cooling; [37]: M. Metcalf, J. E. Moussa, W. A. de Jong and M. Sarovar, Phys. Rev. Research2,023214 (2020), Engineered thermalization and cooling of quantum many-body systems.

---

## Round 3 · Author Response

Dear Editor-in-charge,
We have considered the Referees’ reports concerning our manuscript “Dissipative cooling induced by pulse perturbations”, which we submitted to SciPost for publication. First of all, we thank all three the Referees for their work in reviewing our manuscript. That being said, we note that, out of the three Referees, both Referee 1 and Referee 3, during the first round, recognize the validity of the results contained in our manuscript while contributed report of Referee 2 arises a list of remarks:
Referee 1:
-
“The manuscript convincingly presents an interesting result […].”
-
“The mechanism studied in this work could provide an understanding of laser-induced superconductivity.”
Referee 3:
- “Interesting original ideas.”
During the second round, the Referee 1 (that, looking at the reports, we assume to be the Referee 3 of the first round) still agree on the validity of the results (“I still think that the actual results might warrant publication”), Referee 2 accepted our responses to his remarks and Referee 3 (ex Referee 1) agrees on publication (“I think the resubmission satisfactorily addresses all previous concerns and the manuscript could be published in SciPost Physics.”). However, while during the first round the Referee 1 assert that “The manuscript is very accessible and well-written.”, during the second round Referee 2 and Referee 1 (ex Referee 3) still arise some observation on the way the results are presented.
Below, we provide a detailed answer to the Referees’ remarks, together with a list of the major modifications we correspondingly made to the paper to meet Referees’ criticisms.
We thank you in advance for the kind attention in our submission.
Best Regards Andrea Nava (on behalf of all the authors)
Answers to: Anonymous Report 1 on 2021-9-27 (Invited Report)
[Q1] the question of referee two why the laser-interaction appears as an interaction between two spins is explained in the response of the referees but not in the manuscript
[A1] We further improved the answer to Referee 2, implementing her/his further observations around the laser-interaction Hamiltonian and adding to the manuscript:
[page 6] “It is worth to stress that, in our model spins and, equivalently, hard core bosons, are just a tool to reproduce a Hilbert space comprising four energy states for each site, two, $\ket{0;i}$ and $\ket{1;i}$, with lower energy and two, $\ket{2;i}$ and $\ket{3;i}$, with higher one. The laser pulse allows transitions between the two sectors, specifically, $0 \leftrightarrow 3$ and $1 \leftrightarrow 2$, which are simply interpreted in terms of hard-core bosons. The key to our mechanism is to tune the laser frequency in resonance with $1 \leftrightarrow 2$, thus depleting the occupation of boson $b_{1,i}$ and increasing that of $b_{2,i}$. In this sense (19) mimics a laser pulse, acting at the level of single particles. In analogy with optical absorption of light we can image that $b_{1,i}$ is even under parity and $b_{2,i}$ odd, so that all transition processes $0 \leftrightarrow 3$ and $1 \leftrightarrow 2$ are dipole active, i.e. couple opposite parity states.”
[Q2] While I still think that the actual results might warrant publication, I feel that the explanations are substantially too scarce to make the manuscript reasonably accessible. […] Fig. 4 for example shows several complicated functions, but there is no explanation of what features in the functions are relevant. […] I don't think that many readers will be able to appreciate the results.
[A2] We thank the Referee for her/his suggestion. We believe that, after the further improvements we made in revising our paper, it is now more accessible and clearer. In the resubmitted version of the manuscript, we extended the explanation and discussion of the results to make them more detailed. All the additions are listed in the “List of major changes” section of the resubmission letter.
Answers to: Anonymous Report 2 on 2021-10-12 (Contributed Report)
[Q1] I still don't think that introducing the notion "hard core bosons" is very helpful, but I leave the choice with the authors. I would just ask the authors to please write a few more details or at least mention that this is merely a formal construction. Just introducing the bosons by writing "In other words, we can write..." is not helpful. Also, please use the bosons consistently; e.g. below Eq. (19) you resort to projectors/ket-bra symbols, that could all be represented by bosonic operators for consistency. The same for the jump operators; those could also be expressed in terms of hard core bosons, right?
[A1] We thank the referee for her/his observation. We agree that it would be better to give some more details on the mathematical aspect of the hard core bosons. For this reason, we added to the manuscript, before “the optical analogy”:
[page 6] “It is worth to stress that, in our model spins and, equivalently, hard core bosons, are just a tool to reproduce a Hilbert space comprising four energy states for each site, two, $\ket{0;i}$ and $\ket{1;i}$, with lower energy and two, $\ket{2;i}$ and $\ket{3;i}$, with higher one. The laser pulse allows transitions between the two sectors, specifically, $0 \leftrightarrow 3$ and $1 \leftrightarrow 2$, which are simply interpreted in terms of hard-core bosons. The key to our mechanism is to tune the laser frequency in resonance with $1 \leftrightarrow 2$, thus depleting the occupation of boson $b_{1,i}$ and increasing that of $b_{2,i}$. In this sense (19) mimics a laser pulse, acting at the level of single particles. In analogy with optical absorption of light we can image that $b_{1,i}$ is even under parity and $b_{2,i}$ odd, so that all transition processes $0 \leftrightarrow 3$ and $1 \leftrightarrow 2$ are dipole active, i.e. couple opposite parity states.”
For what concern the consistency related to the hard core bosonic notation, the projectors/ket-bra symbols below Eq.(19) are already expressed in terms of b_1 and b_2 while the Lindblad operators in Eq.(25) can be easily expressed in terms of b_1 and b_2 resorting to Eq.(16). While we find that it would not be helpful to list all the Lindblad operator, one by one, in terms of the hard core bosons, we agree that it is useful to mention it. For this reason, we added, before “Detailed balance”:
[page 8] “Lindblad operators can be easily expressed in terms of hard core bosons through Eq.(16).”
[Q2] Thanks for the clarification. Please rework the sentence that you added to the manuscript. It is not clear what you mean by "optical analogy" (the reader does not know about our exchange here).
[A2] We thank the referee for the suggestion. In [A1] we also rephrased the paragraph regarding the optical analogy including part of our exchange.
[Q3] Can you please spell out what "compatibility with the mean-field character" precisely means?
[A3] In order to make clear what we mean by “compatibility with mean-field character of the Hamiltonian”, we added the following note in the manuscript:
“After resorting to the mean field approximation (let us remember that in our case it is exact in the thermodynamic limit), the Hamiltonian becomes a function of the density matrix. Indeed, the Hamiltonian parameters (the magnetic field in our case) should be determined by self-consistency conditions like in Eq.(23). It follows that the Hamiltonian eigenvalues becomes a function of the density matrix too. Now, as the set of Lindblad operators is written in terms of the Hamiltonian eigenstates, when a mean field approximation is performed the Lindblad equation becomes a non-linear differential equation with Lindblad operators that are implicitly a function of the density matrix. Furthermore, in principle a Lindblad operator could connect any two states within the Hilbert space of an interacting Hamiltonian but, after the mean field approximation, as the density matrix factorises into the product of single-site density matrices, the Lindblad operators only involves eigenstates at the same site. See Refs.[26, 39, 40] for some examples.”
[39] PHYSICAL REVIEW B 100, 125102 (2019), Andrea Nava and Michele Fabrizio, Lindblad dissipative dynamics in the presence of phase coexistence
[40] PHYSICAL REVIEW B 103, 115139 (2021), Andrea Nava , Marco Rossi , and Domenico Giuliano, Lindblad equation approach to the determination of the optimal working point in nonequilibrium stationary states of an interacting electronic one-dimensional system: Application to the spinless Hubbard chain in the clean and in the weakly disordered limit
[Q4] Thank you, I accept that. But could you please explain in what sense the strength of the coupling to the laser pulse is the "most critical parameter"? Also, rather than writing "we make some not unphysical assumptions", you could just spell out once more what assumptions you are making. And after that you can re-iterate why they are reasonable/physical.
[A4] In order to highlight the discussion about the unphysical assumptions and the strength of the coupling to the laser pulse, we added into the conclusions of the manuscript
“It is worth stressing that, while we make some not unphysical assumptions on the dissipative processes, we also show that the selective cooling persists for a wide range of parameters. Indeed, we just assume that high-energy excitations dissipate faster than low-energy ones, which is a rather generic circumstance. In reality, the most critical parameter here is the strength of the coupling to the ‘laser pulse’, i.e., the dipole matrix element in real materials, or, equivalently, the intensity of the optical absorption that is related to the strength of the transition processes. In our toy model there is just a single transition process that connects the low energy sector, states 0 and 1, with the high energy one, states 2 and 3. Evidently, the cooling is less efficient the smaller the coupling strength. However, regarding K3C60, we recall that in the experiment the laser hits a rather pronounced mid-infrared peak, which we have associated to an exciton that plays the role of the high-energy sector [20], while the role of the low energy sector is there played by particle-hole excitations. The intensity of that peak reflects the substantial strength of the process. It is self-evident that it would be totally useless to pump at an excitation with very small absorption strength. Indeed, the strength of the optical process is a prerequisite of our cooling strategy.”
Answers to: Anonymous Report 3 on 2021-10-13 (Invited Report)
We thank the Referee for her/his appreciation of our work.
[Q1] The expression of the interaction with the laser field in terms of two distinct spins actually makes sense because of the vastly different energy scales being involved. Making this point more explicit in the manuscript might be a good idea, but it is not strictly necessary for the understanding of the paper. […] I also think that the authors comment on Fig. 4 in the main text in an adequate way, especially concerning the influence of r on relaxation time.
[A1] We thank the Referee for her/his suggestion. In the revised version of the manuscript, we extended the discussion at page 6 in order to meet other Referees’ request too and added some more comment about the model and the results here and there in the manuscript (as reported in the list of major changes).

---

## Round 3 · List of Changes

Warnings issued while processing user-supplied markup:
- Inconsistency: plain/Markdown and reStructuredText syntaxes are mixed. Markdown will be used.
Add "#coerce:reST" or "#coerce:plain" as the first line of your text to force reStructuredText or no markup.
You may also contact the helpdesk if the formatting is incorrect and you are unable to edit your text.
List of major changes
[page 6] “It is worth to stress that, in our model spins and, equivalently, hard core bosons, are just a tool to reproduce a Hilbert space comprising four energy states for each site, two, $\ket{0;i}$ and $\ket{1;i}$, with lower energy and two, $\ket{2;i}$ and $\ket{3;i}$, with higher one. The laser pulse allows transitions between the two sectors, specifically, $0 \leftrightarrow 3$ and $1 \leftrightarrow 2$, which are simply interpreted in terms of hard-core bosons. The key to our mechanism is to tune the laser frequency in resonance with $1 \leftrightarrow 2$, thus depleting the occupation of boson $b_{1,i}$ and increasing that of $b_{2,i}$. In this sense \eqn{eq:envelope} mimics a laser pulse, acting at the level of single particles. In analogy with optical absorption of light we can image that $b_{1,i}$ is even under parity and $b_{2,i}$ odd, so that all transition processes $0 \leftrightarrow 3$ and $1 \leftrightarrow 2$ are dipole active, i.e. couple opposite parity states.”
[page 8] “Lindblad operators can be easily expressed in terms of hard core bosons through Eq.(16).”
[page 9] “In Fig. 4 we show the results of the numerical integration of Eq. (24) at temperature T = 1.5 Tc, and for increasing r, see Eq. (27), from r = 1, blue line, to r = 640, purple line. The laser pulse parameters, see Eq. (19), are E0 = 0.2 and τ = 1000. As long as the laser pulse is on, the order parameter time evolution is similar to the non dissipative case shown in Fig. 2, i.e. it starts from $m_{x,1}=0$, corresponding to the disordered system, and evolves toward one of the two Z2-equivalent symmetry ordered phases. At the end of the laser pulse, while in the non dissipative case the system remains trapped in the ordered phase with $m_{x,1}\neq0$, in the dissipative case the systems tends to return in the disordered one after a time that depends on the ratio r. We note that when low-energy excitations dissipate as fast as high-energy ones, blue line at r = 1, the system quickly relax to thermal equilibrium, $m_{x,1}=0$, without showing any transient cooling. The latter appears only upon increasing r, and lasts longer the larger r is. To quantify how long […]”
[page 13] “Again, the result shows an interesting qualitative agreement with the experimental observation reported in panel a) of Fig.5 of Ref. [14] where, indeed, ’the photoresistivity was mostly independent of the pump-pulse duration and depended only on the total energy of the excitation pulse’. Indeed, as observed in Ref.[14], this is a rather interesting aspect as the fact that the amplitude of the effect only depends on the laser pulse fluence is a strong evidence against the early proposed approaches based on the non-linear phonon mechanism where the system response is expected to depend on the laser pulse peak amplitude [12] while it looks to be consistent with the selective cooling mechanism.”
[page 13] “It is worth stressing that, while we make some not unphysical assumptions on the dissipative processes, we also show that the selective cooling persists for a wide range of parameters. Indeed, we just assume that high-energy excitations dissipate faster than low-energy ones, which is a rather generic circumstance. In reality, the most critical parameter here is the strength of the coupling to the ‘laser pulse’, i.e., the dipole matrix element in real materials, or, equivalently, the intensity of the optical absorption that is related to the strength of the transition processes. In our toy model there is just a single transition process that connects the low energy sector, states 0 and 1, with the high energy one, states 2 and 3. Evidently, the cooling is less efficient the smaller the coupling strength. However, regarding K3C60, we recall that in the experiment the laser hits a rather pronounced mid-infrared peak, which we have associated to an exciton that plays the role of the high-energy sector [20], while the role of the low energy sector is there played by particle-hole excitations. The intensity of that peak reflects the substantial strength of the process. It is self-evident that it would be totally useless to pump at an excitation with very small absorption strength. Indeed, the strength of the optical process is a prerequisite of our cooling strategy.”
[note] “After resorting to the mean field approximation (let us remember that in our case it is exact in the thermodynamic limit), the Hamiltonian becomes a function of the density matrix. Indeed, the Hamiltonian parameters (the magnetic field in our case) should be determined by self-consistency conditions like in Eq.(23). It follows that the Hamiltonian eigenvalues becomes a function of the density matrix too. Now, as the set of Lindblad operators is written in terms of the Hamiltonian eigenstates, when a mean field approximation is performed the Lindblad equation becomes a non-linear differential equation with Lindblad operators that are implicitly a function of the density matrix. Furthermore, in principle a Lindblad operator could connect any two states within the Hilbert space of an interacting Hamiltonian but, after the mean field approximation, as the density matrix factorises into the product of single-site density matrices, the Lindblad operators only involves eigenstates at the same site. See Refs.[26, 39, 40] for some examples.”
We added the following references:
[39] PHYSICAL REVIEW B 100, 125102 (2019), Andrea Nava and Michele Fabrizio, Lindblad dissipative dynamics in the presence of phase coexistence
[40] PHYSICAL REVIEW B 103, 115139 (2021), Andrea Nava, Marco Rossi , and Domenico Giuliano, Lindblad equation approach to the determination of the optimal working point in nonequilibrium stationary states of an interacting electronic one-dimensional system: Application to the spinless Hubbard chain in the clean and in the weakly disordered limit

---

## Editorial Decision

published